# A Novel Generative Adversarial Network-Based Approach for Automated Brain Tumour Segmentation

**DOI:** 10.3390/medicina59010119

**Published:** 2023-01-06

**Authors:** Roohi Sille, Tanupriya Choudhury, Ashutosh Sharma, Piyush Chauhan, Ravi Tomar, Durgansh Sharma

**Affiliations:** 1School of Computer Science, University of Petroleum and Energy Studies (UPES), Dehradun 248007, India; 2Persistent Systems, India 411057, India; 3School of Business and Management, CHRIST University, Bangalore 560074, India

**Keywords:** generative adversarial learning, tumour segmentation, brain MRI, deep learning, autoencoder

## Abstract

*Background*: Medical image segmentation is more complicated and demanding than ordinary image segmentation due to the density of medical pictures. A brain tumour is the most common cause of high mortality. *Objectives*: Extraction of tumorous cells is particularly difficult due to the differences between tumorous and non-tumorous cells. In ordinary convolutional neural networks, local background information is restricted. As a result, previous deep learning algorithms in medical imaging have struggled to detect anomalies in diverse cells. *Methods*: As a solution to this challenge, a deep convolutional generative adversarial network for tumour segmentation from brain Magnetic resonance Imaging (MRI) images is proposed. A generator and a discriminator are the two networks that make up the proposed model. This network focuses on tumour localisation, noise-related issues, and social class disparities. *Results*: Dice Score Coefficient (DSC), Peak Signal to Noise Ratio (PSNR), and Structural Index Similarity (SSIM) are all generally 0.894, 62.084 dB, and 0.88912, respectively. The model’s accuracy has improved to 97 percent, and its loss has reduced to 0.012. *Conclusions*: Experiments reveal that the proposed approach may successfully segment tumorous and benign tissues. As a result, a novel brain tumour segmentation approach has been created.

## 1. Introduction

A malignant tumour is an extremely harmful health risk that can be fatal. To reduce the community’s fatality rate, early detection, diagnosis, and treatment are essential. Cancerous cells can now be found using a range of imaging modalities, including magnetic resonance imaging (MRI), computed tomography scans (CT), positron emission tomography (PET), and X-rays, thanks to advancements in medical imaging technology. High resolution, a high signal-to-noise ratio, and the ability to image soft tissues are all advantages that MRI has over other imaging modalities [1]. Compared to MRI images, CT scans offer poorer contrast in soft tissues. For these reasons, MRI is the method that is most frequently used for segmenting and diagnosing brain tumours. Compared to CT pictures, MRI scans show a noticeable contrast between tumorous and non-tumorous cells. Brain MRI is broken down into different parts, such as white matter, cerebrospinal fluid, grey matter, and various lesions/tumours, for the purpose of analysing anomalies in brain pictures. Brain MRI imaging uses the modalities of spin-lattice relaxation (T1-weighted), spin-spin relaxation (T2-weighted), and fluid attenuation intention recovery (FLAIR). Each tissue has a unique indication because of the variations in these modalities [2]. Due to the high contrast value, tumours can be easily separated from normal tissue in MRI scans. Brain MR scans can be used by the radiologist to detect various lesions and cancers, which helps with medication recommendations. Due to the reliability issues with many sensory modalities, segmenting medical images can be challenging. The manual segmentation of tumour cells is a labour-intensive and time-consuming technique. In addition, certain artefacts, such as motion artefacts, have an impact on image segmentation. Partial volume effect (PVE) is caused when healthy tissues overlap in terms of intensity, and PVE-like characteristics can be seen in tumorous tissues [3].

Medical images are affected by the noise from accessories and auxiliary devices. For the aim of making a diagnosis, this method is crucial for extracting data from an image. Effective and precise tumorous zone segregation is essential for brain MRI segmentation. So, brain tumours are divided using automated segmentation methods. In a real-time situation, automatic segmentation will help radiologists make more rapid and accurate diagnoses of cancers. Deep belief networks, restricted Boltzmann machines, stack auto-encoder networks, and deep convolutional neural networks are examples of automated segmentation methods. The most widely used segmentation method in biomedical image processing, convolutional neural networks, allows for more accurate segmentation and identification of brain MRI signals. According to existing methods, algorithm efficiency in the classification and segmentation of tumorous and non-tumorous cells should be enhanced. Despite their effectiveness, deep convolutional neural networks have several limitations in terms of what they can do. Existing computer-assisted diagnosis methods are unreliable because of how inaccurate the trained model is. According to the literature assessment, there are a number of issues with current technology.

Classification and segmentation tasks are not aligned with one another. Models for segmentation and classification must be distinct from one another. Lesions, cancers, and healthy cells could not be distinguished with any accuracy by earlier models. In order to detect tiny lesions as tumorous cells, they are frequently segmented. The class imbalance between tumorous and non-tumorous cells exists in earlier versions, and they are rigid when it comes to adjusting layer sizes for different input sizes across different datasets.

Both benign and malignant brain tumours are possible. Unlike malignant tumours, which are cancerous, benign tumours can be treated because they are not cancerous. If a cure is not found, especially for malignant tumours, the patient may pass away. Because of this, early tumour prediction and detection can help to lower the death rate. At a very early stage, cancers can be found in any picture modality using automated artificial intelligence approaches. For this purpose, real-time segmentation of brain tumours from the different modality scans are required. GAN’s have proven to attain much high efficiency for brain tumour segmentation in a real-time scenario.

Convolutional neural networks (CNN) and generative adversarial networks (GAN), two deep learning approaches, are mostly used for automated brain tumour segmentation. In contrast to GANs, CNNs are hybrid deep learning models that can make decisions based on a variety of inputs. Unlike CNNs, which need huge, labelled datasets for their training, GANs use unsupervised learning and do not need as many large datasets. Fewer labelled datasets provided to GANs during training can shorten training time while simultaneously improving accuracy or efficiency of the network. The following advantages of GAN over CNN are discussed in this work along with an overview of the several GAN-based designs.

Generative adversarial networks are used to enhance the accuracy of currently used computer-assisted technologies. Through unsupervised learning using generative adversarial networks, the basic data distribution from a collection of supplied samples is effectively captured [4]. When working with high-dimensional data, including images and text, this technique becomes more difficult. In order to achieve this, we employ generative adversarial networks [5], which offer a mapping from the latent space to the high-dimensional data. GAN’s capacity to extract information from all types of image data has led to more promising findings in the segmentation of MRI data, such as when using MRI data to segment CT images [6].

Analysing MRI data can be facilitated by computer-aided diagnostics (CAD). The interest in creating CAD-based methods based on deep learning and artificial intelligence has significantly increased recently. Deep learning techniques, however, require training with lots of medical imaging data. GANs, or generative adversarial networks, are capable of creating fresh samples of data and accurately simulating the distribution of the actual data. The generator and discriminator neural networks in GANs, a specific kind of deep learning models, are combined. The discriminator aims to categorise the images as real or artificial while the generator creates fresh examples. The overall training of the model is significantly improved by the adversarial training. In addition to being used for applications such as super-resolution, segmentation, and diagnosis, GANs methods have also been employed for the generation of synthetic data in the field of medical imaging.

The major contributions of this research work are:In order to significantly improve tumour localisation and tumour segmentation, a real-time generative adversarial network is proposed. The generator is generating the segmented tumour output which is compared with the ground truth tumour mask in discriminator section.The model has attained a comparatively high accuracy in segmenting high-resolution images of brain tumour.The exact tumour areas have been clearly marked by the model.

The structure of this essay is as follows: The history of automated brain tumour segmentation techniques and the application of GAN networks are covered in Section 2. Additionally, it discusses related research examining the efficacy, dice score coefficient, and other metrics of tumour segmentation techniques based on GAN. The proposed technique and the associated algorithms are described in Section 3. In Section 4, values obtained from various performance metrics are discussed together with the quantitative and qualitative outcomes of the automated brain tumour segmentation technology that is suggested. Section 5 talks about RTGAN’s conclusion and upcoming work.

## 2. Background and Related Work

The segmentation of brain tumours is being accelerated and automated as a result of extensive research. Because manual brain tumour segmentation is a laborious and time-consuming process, death risk is increased due to the delayed diagnosis of lesions or sick tissues. In order to lower the mortality rate, efforts are being undertaken in the area of brain tumour segmentation automation, which helps the radiologist make an accurate tumour segmentation diagnosis quickly. With the help of very effective deep learning techniques, research is being conducted to develop automated brain tumour segmentation. Because of benefits such as the ones listed below, deep learning algorithms are favoured to alternative techniques:Do not require labelled datasets.Highly efficient.Fast computational speedReal-time diagnosis.Robust.

Generative adversarial networks are the most effective for brain tumour segmentation tasks, according to current study findings. This is because they can effectively handle class imbalance losses and properly distinguish tumorous from non-tumorous tissues in brain MRI images. A generative adversarial network was recommended by Tony C. W. Mok et al. for data augmentation [7]. Authentic data augmentation is necessary because a sizable amount of biomedical data is not available for training. CNNs are not used for this task; instead, GANs are used to collect the training data and produce authentic enhanced data automatically.

Three-dimensional conditional GANs have been proposed by Meen Rezeai et al. to segment brain images and address the problem of class imbalance losses [8]. Because of this, voxels associated with healthy tissues and unhealthy or tumorous tissues may be distinguished with accuracy. For categorising ultrasound pictures as thyroid or healthy tissues, [9] presented dual-path semi-supervised conditional generative adversarial networks.

SeUDA is an unsupervised domain adaptation that was suggested by Chen et al. [10]. This transfers information from one chest X-ray to another image by adapting it. The semantic aware generative adversarial networks are included. Specifically, the target data mapped to the chest X-ray source data as well as the loss limitation in the semantic aware GAN.

According to Jue Jiang et al. [11], a special GAN model has been developed that transforms the data from CT pictures into MRI images. The genuine MRI pictures and the labels from the GAN-generated MRI images were then combined, and the segmentation network was trained. These studies have led to greater accuracy in segmenting malignant tissues with smaller amounts of MRI data. A knowledge transfer-based shape consistent method was developed for coronary artery segmentation (Fei Yu et al.) [12] due to the manual annotation requirement by supervised deep learning algorithms, which is a time-consuming operation. A shape constrained network was proposed in a manner similar to this (Bingnan Luo et al.) [13], which consists of a variational autoencoder GAN that learns the latent space distribution of eye images and the Segnet, but the losses of the Segnet were changed to intersection over union loss, shape discriminator loss, and shape embedding loss.

For forecasting future disorders, such as the prognosis of the transformation of moderate cognitive decline to Alzheimer’s disease in aged MRI pictures, another prediction model (Viktor Wegmayr et al.) [14] proposed consisting of the Wasserstein GAN network was made. The model achieved great levels of precision, recall, and accuracy. Figure 1 shows the flow of work conducted on GAN based models on yearly basis.

Table 1 represents the survey conducted on various GAN based models including parameters like methods used, datasets used, computation time and evaluation parameters. It provides a summary of the literature provided at above.

RescueNet recommended dividing the entire tumour into its core and enhancing regions during a brain MRI scan. This residual cyclic unpaired encoder-decoder employs both residual and mirroring techniques. The proposed network addresses the challenge of labelling large datasets by employing unpaired adversarial training [15]. The dice scores for the entire tumour, the core tumour, and the augmenting tumour are 0.94, 0.85, and 0.93, respectively. Sensitivity values for the entire tumour, the core tumour, and the augmenting tumour are, respectively, 0.91, 0.86, and 0.95.

A parasitic GAN suggested a more effective way to utilise the unlabelled datasets. The segmentor, generator, and discriminator in this suggested network are all functional. With the use of segmentor-produced label maps and generator-synthesised label maps in this network, the discriminator is able to understand the precise periphery of ground truth. As a result, the segmentor can benefit from adversarial learning techniques and extra observation that the discriminator offers. Due to the named relationship between the segmentor, discriminator, and generator, the proposed model was given a name. As a result, segmentor’s fitness ability is constrained, which enhances the capability of the segmentor as a whole [16].

A 3D volume to volume GAN is also recommended for automatically segmenting brain tumours. To get precise segmentation results, this model inputs multi-channel 3D MR images. The BraTS dataset was used to test the model, and it was successful in classifying brain tumours into three categories: whole tumour, active tumour, and enhancing tumour. These categories were distinguished by dice scores of 87.20 percent, 81.14 percent, and 78.67 percent, as well as Hausdorff distance values of 6.44 mm, 24.36 mm, and 18.95 mm, respectively [17].

The output is more accurate and precise when segmenting tumours when GAN is combined with autoencoder learning representation of input data. This model was given the moniker GAN-segNet. Convolutional techniques were used in this to semantically regularise the extracted information. The additional autoencoder aids in adjusting the scales of extracted features such that some insignificant features are omitted. As a result, the system is both dense and light. In order to effectively reduce the impact of label inequity, an additional loss function was included to keep track of loss features. The model was trained and evaluated on BraTS 2018 and produced the following dice scores for the overall tumour, core tumour, and enhancing tumour of 0.9022, 0.814, and 0.8280 [8].

A new GAN design (voxel-GAN) was suggested to help automated BTS deal with the uneven data issue. According to this concept, the majority of voxels are found in healthy areas, while very few are found in lesions, tumours, or diseased tissues. Segmenter and discriminator are both included in this 3D conditional GAN. When the segmentor is trained on MR images, the segmentation labels at the voxel level are learned. A discriminator is then taught to distinguish the segmentor output (ground truth). Both sub-networks have received good training to reduce label discrepancy problems. The suggested model was tested using data from BraTS 2018 and ISLES 2018, demonstrating that it produced meaningful findings [18].

It is suggested to use GAU-Net, a combination of channel and spatial attention mechanisms that incorporate multiple convolutions and are further integrated with U-net. In addition, a residual module for conventional up- and downsampling has been added. The BraTS 2018 dataset was used to train this model, which resulted in a significant improvement in mIoU of 0.75 and a decrease in inference time [19].

It was suggested to combine label refinement and sample reweighting techniques into a single framework for 3D GAN. On the BraTS 2019 dataset, exclusive tests with the suggested model have been conducted. This model achieved competitive performance on managing incorrect labels in automated segmentation of brain tumours when compared to other benchmark models [20].

The SAM-GAN model, which combines local mutual information maximisation and attention mechanisms, has been presented. A semi-supervised model is used. Through the employment of channel and spatial attention blocks, the attention mechanism focuses on what and where to focus. The other technique focuses on local image dependencies in order to increase the network’s capacity to demonstrate itself more consistently. When compared to other fully supervised and semi-supervised networks, SAM-GAN performs better in terms of accuracy and efficiency when segmenting brain tumours [21].

It was suggested as an inductive transfer learning strategy to use unsupervised domain adaptation based on Cycle Gan. This approach to transfer learning helped with the problem of translating annotation labels from source domain datasets to target domain datasets. The use of a transfer learning technique significantly enhanced the semantic segmentation of data related to brain tumours. The suggested method can significantly advance the field of medical image analysis [22] by creating a fundamental instrument to enhance and advance numerous activities involving medical images.

An automated end-to-end network based on Generative Adversarial Nets (GAN) is created for the segmentation of brain tumours, and its accuracy is tested using datasets from BraTS 2015. We propose generative adversarial networks for high-order smoothing in place of conditional random fields. The process of segmenting a single patient’s case has become much quicker, and the results of brain tumour segmentation have improved tremendously [23].

Cascaded GANs with a segmentor that creates a label map and a discriminator that aids in finding the solution while taking into account both short- and long-distance spatial correlations among the pixels were suggested for the whole, core, and enhancing tumour. To achieve a higher dice score coefficient, the proposed model was able to lower the false positives. The model employs a cross-entropy loss (final layer) and a multi-scale loss function (intermediate layer) to achieve improved semantic segmentation efficiency. Furthermore, unnecessary contour smoothing is removed using a multi-scale loss function. The suggested approach outperformed the state-of-the-art strategies for the whole tumour, the tumour core, and the enhancing tumour, respectively, with Dice scores of 0.874, 0.783, and 0.820 [24].

To obtain a non-linear mapping between pictures of the right and left brains, SD-GAN (symmetric driven GAN) was presented. A method for segmenting brain tumours without supervision was proposed in this model. The model’s asymmetry, combined with the fact that it is constructed on upper-level errors, has been trained to replicate ill brains and distinct brain cancers. BraTS 2012 and 2018 were used to train and test the suggested model. Since SD-Gan is unsupervised, it achieved a greater level of accuracy than the benchmark models of CNN and GAN. This study established the validity of the use of unannotated normal MR data to represent symmetric features with underlying structural alterations, and hence their applicability in clinical settings [25].

In order to improve the segmentation outcomes in brain MRI scans, proposed GAN was employed to produce high contrast images. When the CNN model is trained on the output produced by the GAN, the segmentation is greatly enhanced. The number of true channels for segmentation was lowered when these fake images were contrasted with actual images of brain tumour tissue from MR scans. Synthetic images are used as a stand-in for actual channels and are capable of skipping actual modalities in the multimodal brain tumour segmentation framework. Our suggested method is capable of effectively segmenting tumour areas, as evidenced by the results obtained using the BraTS 2019 dataset [26].

This study offers a semi-supervised technique for identifying brain lesions using MRI that makes use of Generative Adversarial Networks (GANs). A generator network and a discriminator network are the two networks that make up a GAN, and they are both trained simultaneously with the intention of enhancing one another. We trained the networks on non-lesion regions from four different MR categorisations. After the network was trained on the BraTS dataset, patches were taken out of areas other than the tumour zone. The underlying probability distribution of the training data is simulated by the generator network to produce data (PData). The discriminator obtains the ensuing probability P by categorising training data and generating data as “Real” or “Fake” (Label Data). After mastering the joint distribution, the generator creates images and patches with arbitrary discriminator performance. During testing, the discriminator assigns posterior probability values for patches from non-lesion regions that are close to 0.5, whereas patches with their centres in lesion sites receive a lower posterior probability value since they are drawn from a different distribution (PLesion). On the test set (n = 14), the proposed technique achieves a whole tumour dice score of 0.69, specificity of 59 percent, and sensitivity of 91 percent. A variety of MRs could be used by the generator network to produce non-lesion patches [27].

Medical image analysis has focused a lot of effort on precisely segmenting the tumour lesions because of the irregularity and blurring of tumour boundaries. This research proposes a brain tumour segmentation method based on generative adversarial networks in light of the current situation (GANs). The GAN architecture is made up of two networks that employ 3D convolutions to combine multi-dimensional context data: a classification network for discrimination and a densely linked three-dimensional (3D) U-Net for segmentation. In order to speed up network convergence and extract more precise data, the densely connected 3D U-Net model incorporates a dense connection. The network can segment several unexpected small tumour subregions thanks to the adversarial training, which brings the distribution of segmentation results closer to that of labelled data [28].

To create artificial MRI images of brain tumours, the AGGrGAN model has been presented. It combines three basic GAN models: two Deep Convolutional Generative Adversarial Network (DCGAN) variations and a Wasserstein GAN (WGAN). To improve the image resemblance, we also used the style transfer technique. By effectively overcoming the constraint of data scarcity, our suggested model is able to comprehend the information variation in various representations of the raw photos. The brain tumour dataset and the Multimodal Brain Tumour Segmentation Challenge (BraTS) 2020 dataset, which are both freely accessible, served as the basis for all of our investigations. The proposed models have attained high SSIM values of 0.57 and 0.62 on above mentioned datasets [29].

## 3. Research Methodology

Medical image segmentation may now be performed precisely in real time because of recent developments in generative adversarial networks. Due to their quick and effective learning capabilities, GANs have become more and more popular.

Deep convolutional GANs based on transfer learning are effective at segmenting semantic brain tumours. For semantic segmentation of medical images, GAN’s are appealing due to the learning process and lower heuristic cost [30]. The Vox2vox model [16] served as inspiration for the suggested concept.

As part of the proposed RTGAN, GANs are trained to segment brain MRI images, and then they are used as feature extractors for supervised tasks using discriminator and generator network segments. Both a generator network and a discriminator network are part of the DCGAN. The segmented tumour picture made from the brain MRI is sent to the generator together with the actual segmented tumour images, and the discriminator processes the results. The discriminator then forecasts the labels for the created output and the true output. Following is the generator’s precise configuration:One 3D image with 4 different modalities: T1, T1gd, T2, and FLAIR.Four 3D convolutional down-sampling blocks having kernel size 4, same padding, stride 2, and leakyRelu activation function. The initial filter set is 64, which is doubled after every convolutional block.Four 3D convolutional residual blocks having kernel size, padding, and activation function the same as mentioned above and stride as 1.Three 3D deconvolutional up-sampling blocks having kernel size = 4, stride = 2, and activation function = Relu.One 3D deconvolutional layer with four filters (background, edema tumour (ED), core tumour (NET), and active tumour (ET), each labelled with 0, 1, 2, and 3).

The detailed configuration of discriminator network is as follows:The 3D image generated from generator network and segmentation ground truth.Four 3D convolutional down-sampling blocks having same configuration as in generator.One 3D convolutional layer with kernel size = 4, filter = 1, stride = 1, and same padding.

## 4. Algorithm

1.Start2.Reshaping image I = > 512 ∗ 512 ∗ 512 -> 128 ∗ 128 ∗ 1283.Generator:I fed to 4 3D convolutional down-sampling blocks -> I‘ (16 ∗ 16 ∗ 16 ∗ 256).I‘ is fed to residual blocks with dropout = 0.2 ->I‘‘ (8 ∗ 8 ∗ 8 ∗ 512).I‘‘ is fed to 3 up-sampling blocks generating I1 (64 ∗ 64 ∗ 64 ∗ 128).I1 is fed to 3D deconvolutional layer with softmax function hence generates segmented image I2 -> 128 ∗ 128 ∗ 128 ∗ 44.Discriminator:I2 + ground truth segmented image -> Four 3D convolutional down-sampling blocks ->I2‘ (8 ∗ 8 ∗ 8 ∗ 512)I2‘ -> one 3D convolutional layer -> I3‘I3‘ -> sigmoid activation function -> Final segmented output I3(8 ∗ 8 ∗ 8 ∗ 1)5.End

Dataset and Preprocessing: The BraTS dataset, which consists of 98 patients and a 3D brain MRI dataset including entire tumours, core tumours, and active tumours, was employed in the training. 73 T1, T2, and FLAIR MRI scans are included in each patient’s dataset folder [31,32,33]. The first step in preparing medical photos is noise removal [34]. Each MRI image is subjected to intensity normalisation during the preprocessing stage, and patch augmentation is used to reduce the model’s memory usage.

## 5. Experiments and Discussions

Losses: The generator loss $G_{L}$ and discriminator loss $D_{L}$ of this GAN-based suggested model are both evaluated. The general dice loss $G_{DL}$ in between the ground truth and generator’s output with a scalar coefficient, i.e., and the discriminator output inaccuracy $L_{1}$ between the ground truth and prediction image, are multiplied by a tensor of ones to obtain the generator loss. α ≥ 0,
$$G_{L}=L_{1}\left[D\left(x,\hat{y}\right),1\right]+\alpha G_{DL}\left(y,\hat{y}\right)\;$$

The discriminator loss is calculated by multiplying the $L_{1}$ error of the discriminator output in between the novel picture and the relevant segmented forecast given by the generator by the error of the discriminator output between the original image and ground truth with a tensor of ones, i.e.,
$$D_{L}=L_{1}\left[D\left(x,y\right),1\right]+L_{1}\left[D\left(x,\hat{y}\right),0\right]\;$$

Training Details: Tensorflow 2.1, the Keras library, and Python 3.7 are used to train the suggested RTGAN model. The model is trained and authorised on sub-volumes of size 128 × 128 × 128 from 98 patients over 100 epochs using batch size 4 on a machine with all necessary libraries.

An overview of the discriminator GAN for several layer types, including conv2d, leaky ReLU, flatten, dropout, and dense, is shown in Table 2. The figure also shows the output variations in parameter for each layer type.

The description of each layer of discriminator GAN is elaborated in Figure 2. Figure 3 represents the architecture of discriminator GAN. It represents various layers added to the discriminator model of GAN which will differentiate the segmented output with the ground truth output. 

Figure 4 depicts a summary of generator GAN for several layer types such as Dense, leaky ReLU, reshape Conv2DT, and Conv2D, as well as the output shape of variation in param for each layer type.

Figure 5 represents the architecture of generator GAN. It represents various layers added to the generator model of GAN which will fetch the tumour regions from the real datasets and passed to all the layers of generator that will deeply segment the tumour regions. The description of each layer is elaborated in Figure 2.

Figure 6 depicts a combined summary for the GAN model, with a total parameter count of 7,815,876 as illustrated by this figure. The number of trainable parameters in this GAN model is 7,686,915; the number of non-trainable parameters is 128,961. 

Figure 7 shows the parameter performance for both the discriminator and the generator summary of the GAN model by representing the generator and discriminator loss at every scan of each epoch.

## 6. Results and Observations

The model has been trained for 100 epochs, and the quality of the segmented images for some epochs is shown in Figure 8. The first part of image is the input image fed to the model. The second part is the maximum segmented region and last part is the final segmented output from the RTGAN model.

The model loss is depicted in Figure 9. The loss diminishes as the number of epochs grows, as seen by the graph of loss vs. epochs in the following figure. In order to reduce the loss to 0.012, testing is carried out across 100 epochs.

The accuracy of the model is displayed in Figure 10 at 97%. The accuracy of the model was calculated in discriminator using segmented truth from the generator minus the ground truth. The accuracy graphs are shown in the following figure, and as the number of epochs rises, accuracy also rises. When the model is put through 100 epochs of testing, the test accuracy rises.

The quantitative findings from the GAN algorithm are presented in this section. Table 3 Image 1 has a Structured Similarity Index (SSIM) value of 0.9021, a Peak Signal-to-Noise Ratio (PSNR) value of 57.30dB, and a Dice Score Coefficient (DSC) value of 0.87. The PSNR value is 69.01 dB, the DSC value is 0.88, and the SSIM value is 0.90110, as shown in Image 2. The PSNR value is 59.32 dB, the DSC value is 0.93, and the SSIM value is 0.8251, as shown in Image 3. The value for the dice score coefficient (DSC) is 0.93, 61.21 dB for the PSNR, and 0.8761 for the SSIM, as shown in Image 4. In Image 5, the SSIM value is 0.9121, the PSNR value is 61.65 dB, and the DSC value is 0.80. Image 6 displays the SSIM value as 0.9561, the PSNR value as 60.23 dB, and the DSC value as 0.94. Image 7 displays the SSIM value as 0.9231, PSNR value as 62.16 dB, and DSC value as 0.90

The last figure of results compares the dice score coefficient of proposed model with previous models. Figure 11 illustrates the comparative analysis of proposed work with previous models.

## 7. Conclusions and Future Scope

According to earlier research, GANs are superior to CNN at segmenting medical images. In this study, it is suggested to segment brain tumours in real time using a GAN-based model. Given their better effectiveness, short processing speeds, ability to diagnose problems in real time, and independence from labelled datasets, GANs are preferable to CNN. GANs are hybrid deep learning models, not discriminative deep learning models such as CNNs. Unlike CNNs, GANs do not need large datasets for training because they are unsupervised learning algorithms. The training time for GANs is decreased while the network’s accuracy and effectiveness are increased when only a few labelled datasets are used in training.

This work proposes RTGAN, which consists of a generator and a discriminator. The generator network includes the dense, leaky ReLU, reshape Conv2DT, and Conv2D in addition to the output form of change in param for each layer type. Conv2d, leaky ReLU, flatten, dropout, and dense are some of the different types of discriminator GAN layers. The output parameter variation forms for each layer type also belong to this group.

The effectiveness of the segmented images was assessed after the model was trained across a number of epochs for 100 epochs at a time. The segmentation quality of the images is good, and PSNR, SSIM, and DSC are used for quantitative analysis. RTGAN has demonstrated its ability to generate high-quality segmentation results for evaluation criteria such as the structural similarity index, dice score coefficient, and peak signal-to-noise ratio. DSC, PSNR, and SSIM are all generally 0.894, 62.084 dB, and 0.88912, respectively. The model’s accuracy has improved to 97 percent, and its loss has reduced to 0.012. The suggested model is ideal for real-time applications due to its high precision and dice scoring coefficient. This work’s drawback is that it needs to be tested using a variety of additional image modalities, including those for low-grade gliomas, glioblastomas, and astrocytomas. The correctness of the final test is then obtained. The work could also be compared to other models based on different parameters related to accuracy such as precision, F1 score, etc.

## Figures and Tables

**Figure 1 medicina-59-00119-f001:**
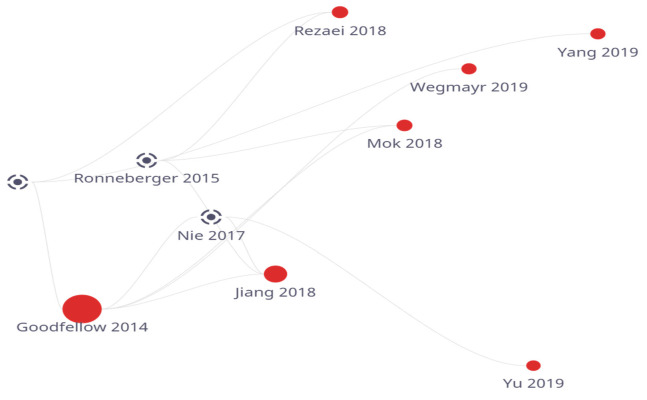
GAN-based model year-wise survey.

**Figure 2 medicina-59-00119-f002:**
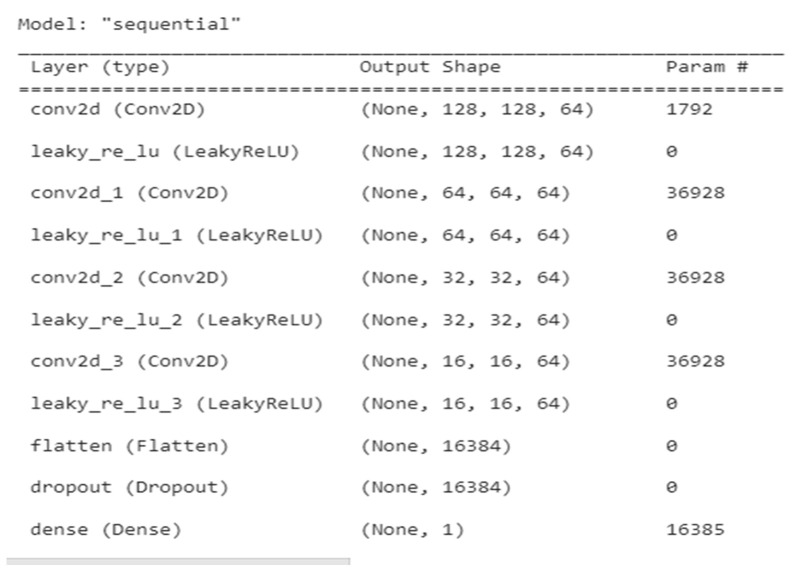
Discriminator summary for tumour segmentation.

**Figure 3 medicina-59-00119-f003:**
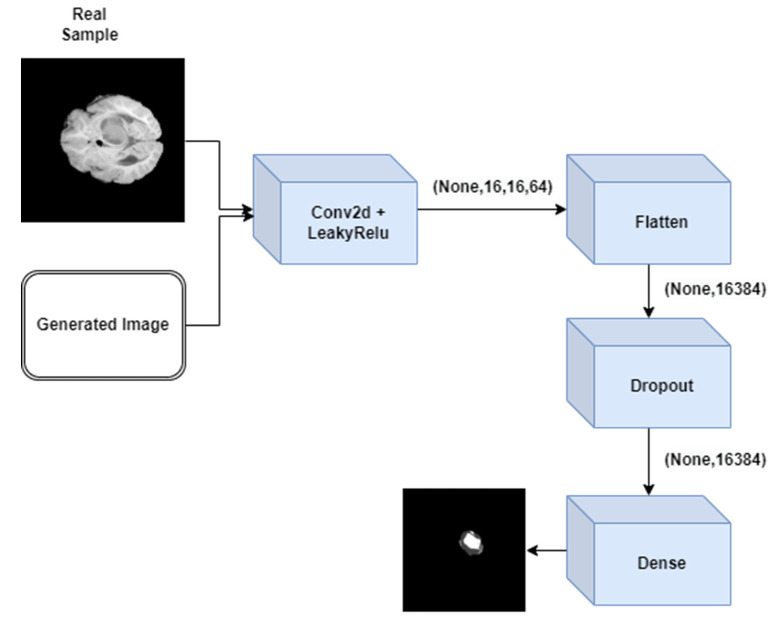
Architecture of discriminator GAN.

**Figure 4 medicina-59-00119-f004:**
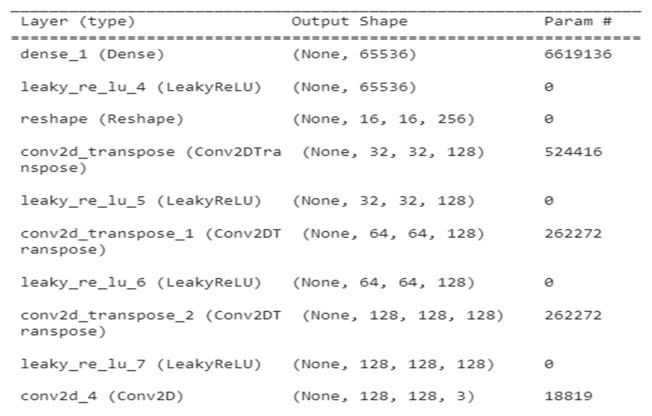
Generator GAN for several layers.

**Figure 5 medicina-59-00119-f005:**
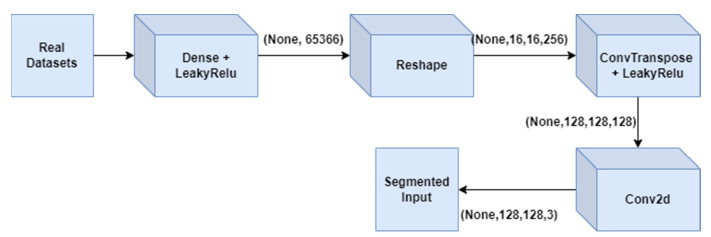
Architecture of generator GAN.

**Figure 6 medicina-59-00119-f006:**
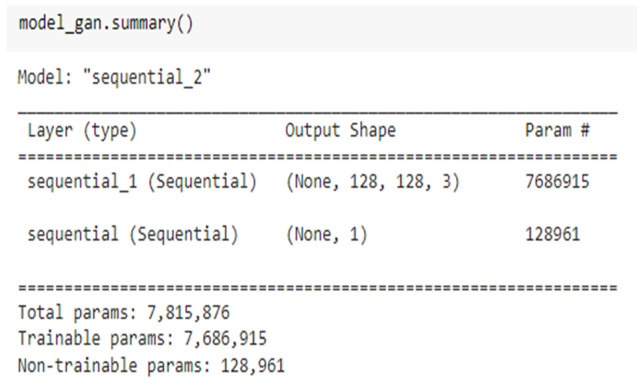
Combined summary for the GAN model.

**Figure 7 medicina-59-00119-f007:**
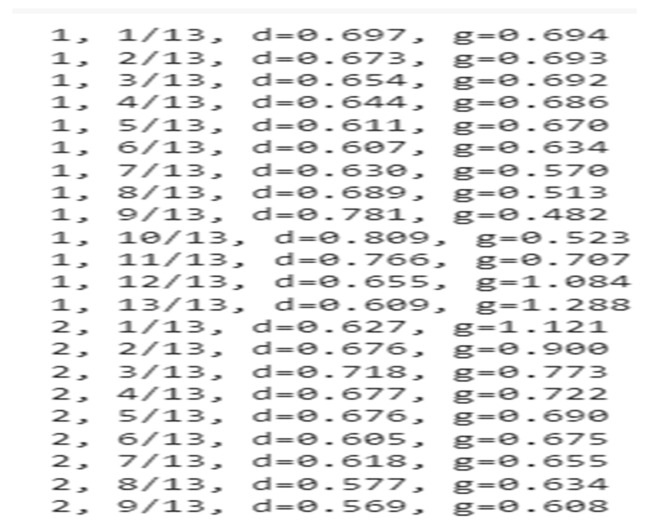
Parameter performance.

**Figure 8 medicina-59-00119-f008:**
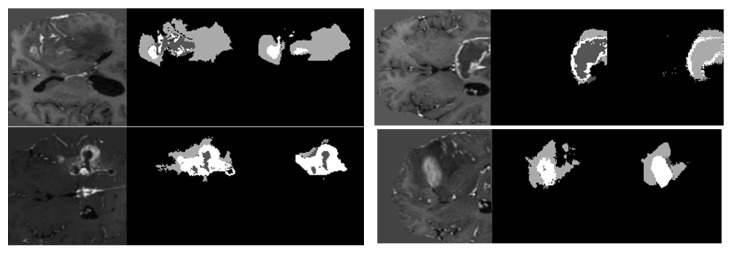
Segmented Output.

**Figure 9 medicina-59-00119-f009:**
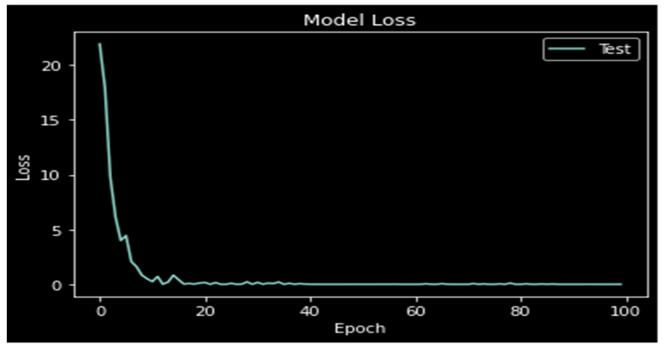
Model Loss.

**Figure 10 medicina-59-00119-f010:**
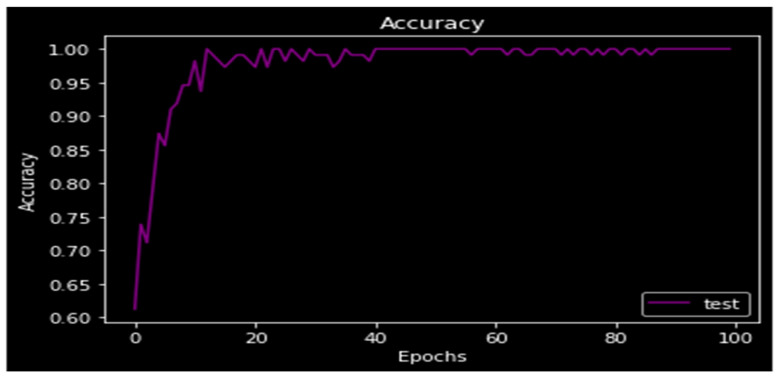
Accuracy.

**Figure 11 medicina-59-00119-f011:**
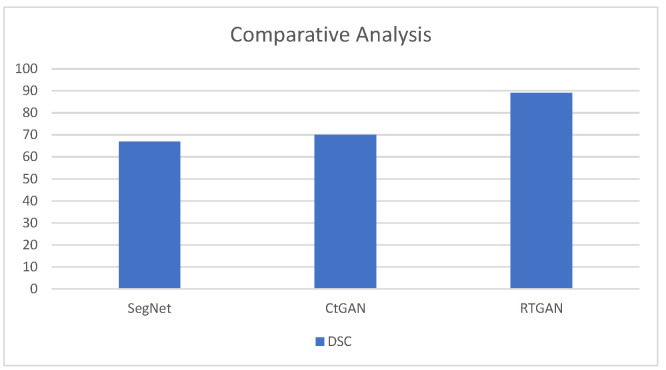
Comparative Analysis of Proposed Work with Previous Models.

**Table 1 medicina-59-00119-t001:** Literature survey of GAN-based models.

Technique	Datasets	Computation Time	Methodology	Evaluation
Tony C.W. Mok et al. [7]	BraTS15	2.1 s for one inference	Generative Adversarial Networks (coarse to fine generators)	DSC: WT = 0.84 CT = 0.63 ET = 0.57
Mina Rezaei et al. [8]	BraTS18		3D Conditional Generative Adversarial Network (cGAN)	DSC (WT = 0.84, CT = 0.79, ET = 0.63) Dice = 0.83 Hausdorf = 9.3 Precision = 0.81 Recall = 0.78
Cheng Chen et al. [10]	JSRT (chest X-ray)		Semantic Aware GAN	DSC = 95.59 Recall-96.59 Precision-94.77
Jue Jiang et al. [11]	T2 MRI images		Tumour Aware loss with GAN	DSC = 0.74
Fei Yu et.al. [12]	DRIVE dataset		Shape-consistent generative adversarial network (SC-GAN)	Accuracy = 0.953 ± 0.009 Precision = 0.820 ± 0.031 Recall = 0.829 ± 0.039 DSC = 0.824 ± 0.026
Bingnan Luo et.al. [13]	Eye Segmentation Dataset (i-bug)	0.033 s	Shape constraint generative adversarial networks	Mean mIOU = 79.02% S-mIOU = 71.86% I-mIOU = 86.185
Viktor Wegmayr et al. [14]	T1 MRI images		Wasserstein-GAN	Accuracy = 73% Precision = 68% Recall = 75%

**Table 2 medicina-59-00119-t002:** Literature Survey of GAN-based Brain tumour Segmentation Models.

Techniques	Datasets	Performance	Advantage	Disadvantage
RescueNet	BraTS 2015 and 2017	Dice = 0.94 Sensitivity = 0.91	Requires less training Data	Not trained on other image modalities.
Parasitic GAN	BraTS 2015 and 2017	Dice score—0.010–0.035	More reliable ground truth for self-taught.	The visual quality of the predicted segmentor should be improved.
3D GAN	BraTS 2020	Dice Score = 87.20% Hausdroff = 6.66	Realistic outputs generated	Model can be improved for different BraTS challenges
GAN-SegNet	BraTS 2018	Dice Score = 0.9022 Positive Predictive Values = 0.9270	Small intratumor region(s) segmentation is improved	Dice score value is not sufficient for real-time use.
Voxel-GAN	BraTS-2018 and ISLES-2018		Mitigates imbalanced data problem in brain tumour semantic segmentation	Dice score value is not sufficient for real-time use.
GAU-Net	BraTS 2018	Increased the mIoU = 0.65 to 0.75 with only 5.4% of U-Net parameters	Captures long-distance dependencies hence improving network performance and less inference time.	Segmentation effect is still far from clinical use.
Cycle GAN	BraTS and ADNI	Dice Score = 0.7086	Can use combination of 2d and 3D data including MRI or CT datasets.	Dice score value is not sufficient for real-time use.
SD-GAN	BraTS 2012 and BraTS 2018	Dice Score = 0.646 Sensitivity = 0.802 Precision = 0.701	Achieved the best-unsupervised segmentation performance	Dice score value is not sufficient for real-time use.

**Table 3 medicina-59-00119-t003:** Results using RT-GAN algorithm.

Images	Image Number	Structured Similarity Index (SSIM)	Peak Signal-to-Noise Ratio (PSNR)	Dice Score Coefficient (DSC)
** 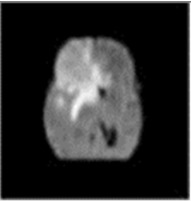 **	I1	0.9021	57.30 dB	0.87
** 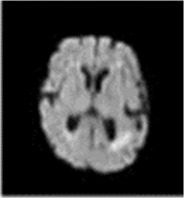 **	I2	0.9110	69.01 dB	0.88
** 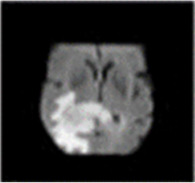 **	I3	0.8251	59.32 dB	0.93
** 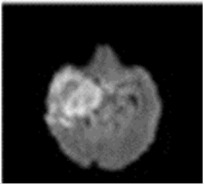 **	I4	0.8761	61.21 dB	0.93
** 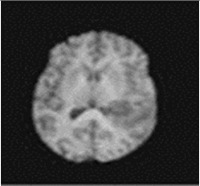 **	I5	0.9121	61.65 dB	0.80
** 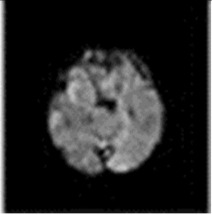 **	I6	0.8366	60.23 dB	0.94
** 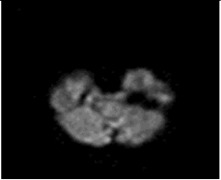 **	I7	0.9231	62.16 dB	0.90
** 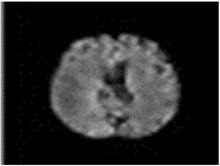 **	I8	0.9081	58.52 dB	0.89
** 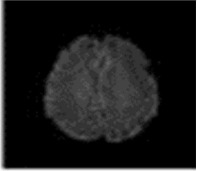 **	I9	0.8845	62.47 dB	0.91
** 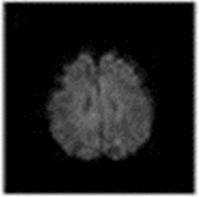 **	I10	0.9125	68.97 dB	0.89

## Data Availability

Datasets are publicly available, and citations related to datasets are mentioned within the paper.

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
