# Peer review of "A Novel Generative Adversarial Network-Based Approach for Automated Brain Tumour Segmentation"

_medicina, 2023, doi:10.3390/medicina59010119_

Round 1

Reviewer 1 Report

Brain tumor is one of the most lethal type of tumor in the world. , Early detection, diagnosis, and treatment are essential for survival. MRI is the method that is most frequently used for segmenting and diagnosing brain tumors. However, the manual analysis of tumor cells is a labor-intensive and time-consuming technique.

Recently, analyzing MRI data can be facilitated by computer-aided diagnostics (CAD). Generative adversarial networks (GAN), one of the deep learning approaches, is mostly used for automated brain tumor segmentation of CAD. 

In this articles, a novel brain tumor segmentation approach (RTGAN) has been created. The authors proved that the method could produce high-quality segmentation results for assessment metrics. The accuracy of the model has increased to 97 percent, while the model loss has decreased. The high accuracy and dice scoring coefficient make it suitable for real-time applications.

Comments:

1.       How did the authors calculate the 97% accuracy rate? Please define the details. Did the authors evaluate the false positive and false negative rate of this new method? Please review the statistics carefully.

2.       Do different brain tumor types (such as low grade glioma, glioblastoma or astrocytoma ) have different accuracy rates? Can this method be used to measure the images of other type of intracranial tumor such as brain metastasis tumor (most from primary lung, brain and liver tumor)? These different type of tumor may have different density character and may need different parameters, or totally different analysis method.

3.       What’s the resolution of these training images and test images? In real world, the images are complex. How will the resolution difference affect the final accuracy rate of this method? Can the authors test and show the accuracy rate of high and low resolution MRI images? 

4.        Do other patient conditions, such as age, gender or race  affect the accuracy rate? 

5.       Several spelling mistakes should be corrected. For example, in line 389, should “param” be “parameter”? In line 449, should “such” be “such as”?

Author Response

Dear Reviewer/Editor,

Thanks.

Reviewer 2 Report

The search is good and needs more improvement.

1. Write the results by numbers in the abstract.

2. Write the most important major contributions at the end of the introduction.

3. Figures 3 and 5 represent the study methodology and should be more clear.

4. Your results should be compared with those of previous relevant studies.

5. What are the limitations you face.

6. Move the "Dataset and Preprocessing:" section of the Results section to a subsection in "3. Research Methodology".

Author Response

Dear Reviewer/Editor,

Thanks.

Round 2

Reviewer 2 Report

Accept in present form